# The Impact of Three Communication Channels on the Dissemination of a Serious Game Designed to Enhance COVID-19 Prevention

**DOI:** 10.3390/ijerph191610143

**Published:** 2022-08-16

**Authors:** Mélanie Suppan, Loric Stuby, Christophe Alain Fehlmann, Mohamed Abbas, Sophia Achab, Stephan Harbarth, Laurent Suppan

**Affiliations:** 1Division of Anesthesiology, Department of Anesthesiology, Clinical Pharmacology, Intensive Care and Emergency Medicine, University of Geneva Hospitals and Faculty of Medicine, 1211 Geneva, Switzerland; 2Genève TEAM Ambulances, 1201 Geneva, Switzerland; 3Division of Emergency Medicine, Department of Anesthesiology, Clinical Pharmacology, Intensive Care and Emergency Medicine, University of Geneva Hospitals and Faculty of Medicine, 1211 Geneva, Switzerland; 4Infection Control Programme and WHO Collaborating Centre on Patient Safety, University of Geneva Hospitals and Faculty of Medicine, 1211 Geneva, Switzerland; 5MRC Centre for Global Infectious Disease Analysis, Imperial College London, London SW7 2BX, UK; 6Specialized Facility in Behavioral Addictions ReConnecte HUG, 1211 Geneva, Switzerland; 7Sociological and Clinical Research Unit SWI-54, WHO Collaborating Center in Training and Research in Mental Health, UniGe, 1211 Geneva, Switzerland

**Keywords:** serious games, infection prevention and control, SARS-CoV-2, COVID-19, infodemiology, communication, public health

## Abstract

Infection prevention interventions can only be effective if they are both well known and easily accessible. A randomized controlled trial showed that a serious game, “Escape COVID-19”, was significantly more effective at improving the intention of adopting adequate infection prevention behavior than regular guidelines among long-term care facility employees. However, less than a fifth of all potential participants were finally recruited in this study. To determine whether a specific communication intervention was more effective than another, we carried out a retrospective analysis of account creation data over a six-month period. During the first period (53 days), information about the serious game was disseminated by a part-time worker. The second period (15 days) corresponded to a press release, while the third period (15 days) reflected an official communication from the Swiss Federal Office of Public Health. A total of 3995 accounts were created during the study period. Most accounts were created by health care workers (2699/3995, 67.6%). Median daily account creation was highest during the press release period (25; Q1:Q3 9:172) and lowest during the official communication period (6; Q1:Q3 4:20). The association between communication intervention and account creation was statistically significant both when considering the overall population (*p* = 0.013) and when only analyzing health care workers (*p* = 0.036).

## 1. Introduction

The COVID-19 pandemic has induced a significant increase in web queries regarding conspiracy theories [1,2]. Likewise, the interest in web hoaxes has been shown to be greater than that in anti-hoax services [3]. Health care workers are also affected [4], and mistrust in health authorities has hampered the dissemination of appropriate infection prevention and control (IPC) guidelines in some regions [5,6,7].

To improve the intention of adopting appropriate IPC behaviors, a serious game, “Escape COVID-19”, was developed using a theory-driven approach [8]. A randomized controlled trial showed that the participants who played the game were significantly more likely to adopt appropriate IPC behaviors than those who were presented with regular IPC guidelines [9]. However, the intended sample size could not be reached and only 295 questionnaires could be analyzed out of the 800 planned (36.9%, 295/800) [10]. Worse, only 652 accounts were created on the platform [9], even though the number of potential participants was estimated to be around 4000 (16.3%, 652/4000) [10]. While some long-term care facilities (LTCFs) had high participation rates, fewer than 10 questionnaires were completed in 77.8% (28/36) of all the LTCFs located in Geneva, Switzerland [9]. The main hypothesis was that even though all LTCFs managers had been sent information regarding this serious game by representatives of the Geneva Directorate of Health, many decided not to forward the information to their employees. The reasons for not forwarding this information are still unclear as the original study was not designed to assess or address this unforeseen issue [10].

Nevertheless, the significant effect this serious game had on the self-reported intention of changing IPC behavior (aOR 3.86, 95% CI 2.18–6.81) [9] induced financial support from the Swiss National Science Foundation (SNSF) through the National Research Program 78 (NRP 78). Since the game was originally developed in French and translated into English by the authors, translations in German and Italian were funded by the SNSF [11]. Furthermore, the SNSF financed a part-time position to help disseminate Escape COVID-19 among health care institutions. This led to the endorsement of this serious game by the Swiss Federal Office of Public Health (FOPH) and by more than 10 independent private and public organizations, such as the Swiss Society for Public Health and the Swiss Red Cross. The whole list can be seen on the official Escape COVID-19 website, which allows anyone to access and play the game freely [12].

Despite this support and the various endorsements, there was no definitive way of ascertaining that Escape COVID-19 would be played by the main target population, i.e., Swiss health care workers (HCWs). Following the hypothesis that different means of disseminating the information regarding the serious game could lead to significant differences in information uptake, the objective of this study was to analyze the activity on the Escape COVID-19 web platform [12] according to three different kinds of communication interventions.

## 2. Materials and Methods

### 2.1. Study Design and Setting

This was a retrospective analysis of prospectively collected data. According to the Swiss federal law on research involving human beings [13], a formal approval from an official ethics committee is not necessary provided that the population cannot be considered as “vulnerable” and as long as no health outcomes are studied. This was confirmed in two prior studies carried out on subsets from the same dataset (Req-2020-01262 and Req-2021-00600) [9,11]. Therefore, we refrained from presenting this specific project to avoid putting an unnecessary burden on our regional ethics committee.

Escape COVID-19 was developed using Articulate Storyline 3 (Articulate Global) [8]. Nicholson’s RECIPE for meaningful gamification [14] and the first 3 steps of the SERES framework [15] guided the design process. The accuracy and reliability of the scientific evidence provided by this game was ensured by the involvement of IPC specialists throughout this iterative development process. Version 3 of the Escape COVID-19 serious game was used throughout the study period (9 March to 8 September 2021).

The design, development, features and data collection and extraction mechanisms of the platform used to host Escape COVID-19 [12] have previously been described [9,10,11]. Briefly, this platform was created under the Joomla! 3.9 content management system (Open Source Matters) [16]. To access the serious game, users were required to create accounts according to the language in which they wished to play the game (French, English, Italian or German) and to whether they were health care professionals, since an abridged version of the game suitable for non-HCWs was available on the platform.

### 2.2. Outcomes

The primary outcome was the number of accounts created on the platform according to specific communication interventions. The secondary outcome was the proportion of accounts created by HCWs during each period. These outcomes were also analyzed considering only accounts created from IP addresses located in Switzerland.

Communication interventions were grouped to form 3 successive and independent periods of dissemination of information on Escape COVID-19:The first period (“SNSF Period”) was defined by the presence of a part-time position funded by the SNSF. This period lasted 53 days, extending from 9 March 2021 (website deployment) to 30 April 2021 (end of funding). During this period, information regarding the Escape COVID-19 serious game was disseminated among health care institutions through this part-time position.The second period (“HUG Period”) corresponded to a press release from the Geneva University Hospitals (HUG) [17]. This press release was published on 3 May 2021 and relayed by many channels, the most important of which was Agence France Presse (AFP) [18]. This period lasted 15 days and extended from 3 May to 17 May 2021, i.e., 5 days after the information was relayed by AFP (12 May 2021).The third period (“FOPH Period”) was linked to the dissemination, by the Swiss FOPH, of information regarding the serious game. On 27 July 2021, a link to the serious game was simultaneously broadcasted on the FOPH’s official website and sent in the NOSO newsletter [19]. On the next day (28 July 2021), the same link was sent with the FOPH COVID-19 newsletter [20]. To be consistent with the timeframe used to analyze the second communication intervention, this third and last period also lasted 15 days (27 July to 10 August 2021).

### 2.3. Data Extraction and Statistical Analysis

Data were extracted to a Microsoft Excel Open Extensible Markup Language (XML) spreadsheet (XLSX) file using the Membership Pro component (version 2.2; Joomla Extensions by JoomDonation) and imported for curation in Stata (version 17; StataCorp LLC). Since both the website and the Membership Pro component were compliant with the General Data Protection Regulation (GDPR) statement, this file did not contain any data generated by users who finally chose to delete their accounts. This could have led to a slight underestimation of the actual number of accounts created on the platform, but there is little reason to believe that this should have changed significantly according to the kind of communication intervention used. Furthermore, a previous study showed that the proportion of users who decided to delete their accounts was relatively low (37/3227, 1.15%) [11].

The chi-squared test was used to compare binary outcomes. The number of accounts created per day, during each period, was reported as median (Q1:Q3). Since variances were significantly unequal according to Bartlett’s test, the Kruskal–Wallis test was used to assess differences in the median number of accounts created during each period. A *p* value < 0.05 was considered significant.

## 3. Results

A total of 3995 accounts were created between 9 March and 8 September 2021. This represented a median of 10 (5:22) accounts created each day throughout the study period. Most accounts were created by HCWs (2699/3995, 67.6%). Registrants resided in 53 different countries. The vast majority were from Switzerland (3231/3995, 80.9%), Germany (336/3995, 8.4%), France (210/3995, 5.3%) and Taiwan (52/3995, 1.3%).

Figure 1 shows the overall number of accounts created per date, and Figure 2 shows a box plot detailing the number of accounts created per day during each study period.

During the first period (SNSF period), a total of 1573 accounts were created, with a median of 21 (12:42) accounts per day. There were significantly more accounts created by HCWs during this period than during the rest of the study (1270/1573, 80.7% vs. 1429/2422, 59.0%, *p* < 0.001). Taking into account only IP addresses located in Switzerland, 1487 accounts were created (1483/1573, 94.3%), with a median of 20 (12:37) accounts created per day. There was no change in the direction or magnitude of the effect regarding the proportion of HCWs who created accounts during this period in comparison to the whole study period (1199/1487, 80.6% vs. 953/1744, 54.6%, *p* < 0.001).

During the second period (HUG period), 1254 accounts were created on the platform, with a median of 25 (9:172) accounts per day. There were significantly fewer accounts created by HCWs during this period (476/1254, 38.0% vs. 2223/2741, 81.1%, *p* < 0.001). Taking into account only IP addresses located in Switzerland, 1134 accounts were created during this period (1134/1254, 90.4%), yielding a median of 12 (5:170). No change was seen regarding the proportion of HCWs who created accounts during this period (431/1134, 38.0% vs. 1′721/2097, 82.1%, *p* < 0.001).

During the third and last period (FOPH period), only 249 accounts were created, with a median of 6 (4:20) accounts per day. Most accounts were created by HCWs (216/249, 87%). During this last period, 228 accounts (228/249, 91.6%) were created from IP addresses located in Switzerland, with a median of 5 (3:20). The proportion of HCWs who created accounts using Swiss IP addresses was similar (200/228, 87.7%).

The overall difference in the median number of accounts created per day was significantly different between the three periods (*p* = 0.013). This difference was similar when taking into account only accounts created from IP addresses located in Switzerland (*p* = 0.015). This difference was also significant when only analyzing HCWs (*p* = 0.036) and even more so when only Swiss IP addresses were taken into account (*p* = 0.027).

## 4. Discussion

### 4.1. Main Considerations

This analysis shows that different communication modalities should be used to disseminate critical IPC information. Indeed, while direct communication with different health care institutions is time consuming, the spike plot shows several peaks in the number of account creations during the first (SNSF) period. Each peak probably represents the effect of a specific communication activity. This theory is in any case supported by the high number of accounts created by HCWs during this first period and by the progressive endorsement of the Escape COVID-19 serious game by different health care associations and institutions.

While somewhat effective, direct communication with health care institutions was nevertheless unable to reach many HCWs. This is probably mostly due to the fact that many HCWs are not part of large institutions and contacting these professionals individually would take much more time than a single part-time position could allow. Therefore, other communication modalities could help reach these professionals more effectively. Indeed, even though the HUG press release generated more interest in the serious game among non-HCWs than among HCWs, the absolute number of HCWs who registered on the platform in the wake of this communication was more than twice as high as the number of HCWs who registered after the FOPH communication.

Although the FOPH website is freely available and easily accessible, and even though anyone, HCWs and non-HCWs alike, can register to receive FOPH newsletters, the communication emanating from this official institution was least effective of all. Three main theories could explain this finding. First, the FOPH communication regarding the Escape COVID-19 serious game was held more than three months after the HUG press release and more than five months after the platform was available online. This represents a long time lag given the rapidly changing landscape of the COVID-19 pandemic. Second, the ever-changing and often conflicting guidelines issued by official institutions have generated a certain degree of distrust [21], and potential participants might not have been keen on following a serious game endorsed by the FOPH. Finally, this communication was held at the very end of a pandemic wave, and interest in IPC measures may have waned between these study periods.

### 4.2. Limitations

The main limitation of the present study is linked to its retrospective design. Indeed, since an analysis of the impact of specific communication channels on account creation was not initially planned, participants were not asked by which means they had heard of the serious game. Therefore, it is impossible to uncontrovertibly claim that the different account creation rates reported here are truly the consequence of the specific communication interventions described in the methods. Nevertheless, the spike plot shows a clear correlation between interventions and account creation, thereby supporting the presence of an association between communication intervention and account creation. The interpretation of the results outlined above is further limited by the fact that a certain degree of overlap between interventions cannot be ruled out. In addition, it was not possible to distinguish specific communication interventions that were conducted during the first study period (SNSF period). Finally, it should be acknowledged that the impact of the different phases of the pandemic waves was not taken into account in the analysis.

### 4.3. Perspectives

Timely dissemination of critical IPC information remains critical to face public health emergencies. Henceforth, health authorities should carefully consider the means of disseminating such information among HCWs and non-HCWs [22]. They should therefore assess the impact of specific communication modalities and already prepare appropriate communication interventions to limit the impact of future crises. Among such modalities, the use of social networks to disseminate critical public health information deserves particular attention [23].

Regarding the serious game itself, many improvements could be considered. Since other viral pandemics may appear unexpectedly, a generic and easily adaptable version of this serious game could be developed to enable a faster response in the case of a new outbreak. Such a version should include all potentially relevant pieces of personal protective equipment. It must also be easily customizable to allow for the distribution of different versions to accommodate cultural and linguistic differences [11]. Furthermore, in the context of the current COVID-19 pandemic, additional modules should be developed to promote vaccination. Such modules should target HCWs and lay people alike.

## 5. Conclusions

In this study, different communication modalities had varying impacts in reaching the intended target population. The use of multiple communication channels should therefore be considered to disseminate critical infection prevention and public health information and guidelines in a timely manner. In addition, official institutions should strive to enhance the effectiveness of their communication and to regain the public’s trust, which seems to have been shaken during the COVID-19 pandemic.

## Figures and Tables

**Figure 1 ijerph-19-10143-f001:**
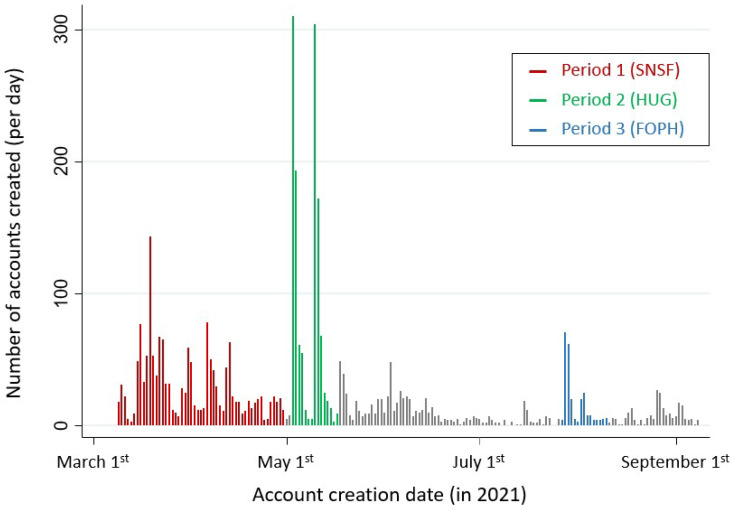
Number of accounts created per date.

**Figure 2 ijerph-19-10143-f002:**
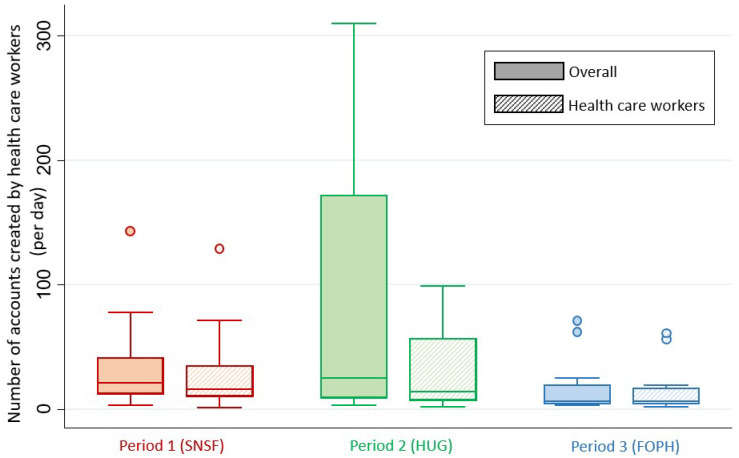
Box plot detailing the number of accounts created per day during each period. SNSF: Swiss National Science Foundation; HUG: Geneva University Hospitals; FOPH: Federal Office of Public Health.

## Data Availability

The data presented in this study can be downloaded from Mendeley Data (doi: 10.17632/3wr6jh496m.1).

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
