# Peer review of "The Impact of Three Communication Channels on the Dissemination of a Serious Game Designed to Enhance COVID-19 Prevention"

_ijerph, 2022, doi:10.3390/ijerph191610143_

Round 1

Reviewer 1 Report

The authors have developed a serious game to enhance COVID-19 prevention in long care facilities. They have published about the development, the protocol and results of a study to the impact of the game. In this manuscript they analyse the effect of three different kinds of communication interventions on the number of new accounts on the web platform of the serious game.

Major comments

1.       People from all over the world signed in for the serious game (lines 135-137). In most cases, communication interventions are targeted within one country, and thus one would like to know what are the best ways of reaching your target groups. Therefore, I would suggest to analyse the data (also) for only the Swiss accounts to exclude possible bias by participants from other countries.

Minor comments

2.       In the introduction, several times ‘answer sets’ are used. What is meant by that? Questionnaires?

3.       LTCF (line 51) and HCW (line 71) are often used abbreviations. Nevertheless, I would suggest to describe it once, before using the abbreviation.

4.       The secondary outcome is now delineated in lines 113-114. I would suggest to add this to the sentences about the primary outcome (lines 92-95).

5.       Lines 214-216: In addition,….COVID-19 pandemic. This sentence does not read correctly, please check.

Author Response

Dear Reviewer,

Thank you very much for your kind and thoughtful comments which have helped us improve the quality of our manuscript. Here are the responses to your comments:

Major Comment 1: People from all over the world signed in for the serious game (lines 135-137). In most cases, communication interventions are targeted within one country, and thus one would like to know what are the best ways of reaching your target groups. Therefore, I would suggest to analyse the data (also) for only the Swiss accounts to exclude possible bias by participants from other countries.
Response: You are of course perfectly right. We have carried out this statistical analysis and the following changes have been made to our manuscript:

  • Methods: we added a sentence to the "Outcomes" subsection: "These outcomes were also analyzed considering only accounts created from IP ad-dresses located in Switzerland."
  • Results: the following sentences have been added:
    • Taking into account only IP addresses located in Switzerland, 1’487 accounts were cre-ated (1’483/1’573, 94.3%), with a median of 20 (12:37) accounts created per day. There was no change in the direction or magnitude of the effect regarding the proportion of HCWs who created accounts during this period in comparison to the whole study pe-riod (1’199/1’487, 80.6% vs 953/1’744, 54.6%, P<.001).
    • Taking into account only IP addresses located in Switzerland, 1’134 accounts were cre-ated during this period (1’134/1’254, 90.4%), yielding a median of 12 (5:170). No change was seen regarding the proportion of HCWs who created accounts during this period (431/1’134, 38.0% vs 1’721/2’097, 82.1%, P<.001).
    • During this last period, 228 accounts (228/249, 91.6%) were created from IP addresses located in Switzerland, with a median of 5 (3:20). The proportion of HCWs who created accounts using Swiss IP addresses was similar (200/228, 87.7%).
    • This difference was similar when taking into account only accounts created from IP addresses located in Switzerland (P=.015). This difference was also significant when only analyzing HCWs (P=0.036), and even more so when only Swiss IP addresses were taken into account (P=.027).

Minor Comment 2: In the introduction, several times ‘answer sets’ are used. What is meant by that? Questionnaires?
Response: This was indeed quite unclear. "Answer sets" were replaced by "questionnaires" in the introduction.

Minor Comment 3: LTCF (line 51) and HCW (line 71) are often used abbreviations. Nevertheless, I would suggest to describe it once, before using the abbreviation.
Response: We apologize for this oversight. These abbreviations have now been defined in the core text.

Minor Comment 4: The secondary outcome is now delineated in lines 113-114. I would suggest to add this to the sentences about the primary outcome (lines 92-95).
Response: We have moved the sentence(s) regarding the secondary outcome as per your comment. The beginning of the "Outcomes" subsection now reads: "The primary outcome was the number of accounts created on the platform ac-cording to specific communication interventions. The secondary outcome was the proportion of accounts created by HCWs during each period. These outcomes were al-so analyzed considering only accounts created from IP addresses located in Switzerland.
Communication interventions were grouped to form 3 successive and independ-ent periods of dissemination of information on Escape Covid-19:"

Minor Comment 5: Lines 214-216: In addition,….COVID-19 pandemic. This sentence does not read correctly, please check.
Response: Thank you for spotting this mistake, a "to" was added to the sentence, which now reads: "In addition, official institutions should strive to enhance the effectiveness of their communication and to regain the public's trust, which seems to have been shaken during the COVID-19 pandemic."

Reviewer 2 Report

It is appropriate for the authors to better explain the characteristics, results and prospects of the spread of Escape game COVID-19. This is what was missing when reading the article.

Thank you!

Author Response

Dear Reviewer,

Thank you very much for your comments and for your suggestions. All your suggestions have been taken into account. The modifications we have carried out are outlined below.

Comment 1: in point 2.1, it is necessary to clearly explain the essence of the proposed game, in particular, build the following table.
Response: The whole development process has been described in a prior publication (ref #8 - Suppan M et al. Development of Escape COVID-19, a Serious Game Designed to Promote Safe Behaviors Among Healthcare Workers During the COVID-19 Pandemic. JMIR Serious Games 2020, doi:10.2196/24986.). Since this was intended to be a brief report, we refrained from adding specific details regarding the development and features of this serious game.
However, your comment made us realize that readers might want to be provided with a brief overview of the serious game. Therefore, the following paragraph was added at the end of the "Study Design and Setting" subsection:
"Escape COVID-19 was developed using Articulate Storyline 3 (Articulate Global). Nicholson's RECIPE for meaningful gamification [REF] and the SERES framework [REF] guided the design process. The accuracy and reliability of the scientific evidence provided by this game was ensured by the involvement of IPC specialists throughout the process."
The following references were added:
- Nicholson, S. A recipe for meaningful gamification. In Gamification in Education and Business; Springer International Publishing, 2015; pp. 1–20 ISBN 9783319102085
- Verschueren, S.; van Aalst, J.; Bangels, A.M.; Toelen, J.; Allegaert, K.; Buffel, C.; Stichele, G. Vander Development of CLInIPUp, a serious game aimed at reducing perioperative anxiety and pain in children: Mixed methods study. J. Med. Internet Res. 2019, 21, doi:10.2196/12429

Comment 2: in point 3, it is appropriate to systematically summarize the obtained results, in particular to build the following table
Response: This is an excellent suggestion that unfortunately cannot be applied to our study. Indeed, this was a single implementation period, which was subdivided post-hoc into 3 communication periods. The retrospective design of this study was acknowledged at the very beginning of the "Study Design and Setting" subsection, but we did not report that the same version of the game was used throughout the study period. In line with your suggestion and with our response, we have now added the following sentence: "Version 3 of the Escape COVID-19 serious game was used throughout the study period (March 9th to September 8th, 2021)."  

Comment 3: in point 4.1, it is appropriate to highlight the prospects for further implementation and improvement of the game, in particular to build the following table or figure:
Response: Thank you for this suggestion. Many of these prospects have been outlined in our prior publications (ref #9 - Suppan M et al. Impact of a Serious Game (Escape COVID-19) on the Intention to Change COVID-19 Control Practices Among Employees of Long-term Care Facilities: Web-Based Randomized Controlled Trial. J. Med. Internet Res. 2021, doi:10.2196/27443 & ref #11 - Suppan M et al. Nationwide Deployment of a Serious Game Designed to Improve COVID-19 Infection Prevention Practices in Switzerland: Prospective Web-Based Study. JMIR Serious Games 2021, doi:10.2196/33003), and this manuscript was only focused on communication aspects regarding the dissemination of this serious game (these aspects should of course be most important for other, future IPC communication interventions). In line with your comment, the following paragraph has now been added to the "Perspectives" subsection:
"Regarding the serious game itself, many improvements could be considered. Since other viral pandemics may appear unexpectedly, a generic and easily adaptable version of this serious game could be developed to enable a faster response in case of a new outbreak. Such a version should include all potentially relevant pieces of personal protective equipment. It must also be easily customizable to allow for the distribution of different versions to accommodate cultural and linguistic differences [11]. Furthermore, in the context of the current COVID-19 pandemic, additional modules should be developed to promote vaccination. Such modules should target HCWs and lay people alike.

Reviewer 3 Report

Interesting topic and well planed. It could be a chance to integrate your discussion with several international study on the topic (use of multichannel to improve communication and disseminate infos in public health domain) so that you can compair your results with other 

Also the biblio needs to be implemented if possible

Author Response

Dear Reviewer,

Thank you for your comments. Here are the adaptations we carried out:

Comment: Interesting topic and well planed. It could be a chance to integrate your discussion with several international study on the topic (use of multichannel to improve communication and disseminate infos in public health domain) so that you can compair your results with other. 
Also the biblio needs to be implemented if possible
Response: Thank you for this insight. Since this was intended only as a brief report and not as a full-fledged article, we decided to refrain from extensively discussing this point. Nevertheless, your comment is important and providing the reader with further references is clearly necessary. Therefore, in line with your comment, we added the following sentence: "Among such modalities, the use of social networks to disseminate critical public health information deserves particular attention."
Along with the following references:
- Ma R, Deng Z, Wu M. Effects of Health Information Dissemination on User Follows and Likes during COVID-19 Outbreak in China: Data and Content Analysis. Int J Environ Res Public Health. 2020 Jul 14;17(14):5081. doi: 10.3390/ijerph17145081. PMID: 32674510; PMCID: PMC7399940.
- McCormack L, Sheridan S, Lewis M, Boudewyns V, Melvin CL, Kistler C, Lux LJ, Cullen K, Lohr KN. Communication and dissemination strategies to facilitate the use of health-related evidence. Evid Rep Technol Assess (Full Rep). 2013 Nov;(213):1-520. doi: 10.23970/ahrqepcerta213. PMID: 24423078; PMCID: PMC4781094.

Round 2

Reviewer 2 Report

It is appropriate for the authors to once again familiarize themselves with the previous review and pay more attention to individual comments and treat them with greater responsibility.

According to the reviewer's suggestions, it is necessary to clearly summarize the research results, which allows to increase the practical value of the article.

Thank you!

Author Response

Dear Reviewer,

Thank you very much for the time you spent carrying out this re-review of our manuscript.

We are sorry that you did not find our responses to be adequate and were deeply saddened to discover that you called our responsibility into question.

We regret that we cannot agree on the modifications which should be made to this manuscript since the modifications you have asked for are not consistent with our objective.

Indeed, the objective of our study is clearly stated at the end of the introduction: "the objective of this study was to analyze the activity on the Escape COVID-19 web platform [12] according to 3 different kinds of communication interventions". Our outcomes are consistent with this objective: "The primary outcome was the number of accounts created on the platform ac-cording to specific communication interventions. The secondary outcome was the proportion of accounts created by HCWs during each period.".

Since the information you have asked for have been included in prior publications, we decided to create a "brief report" (usually < 2'500 words) rather than a full article and referenced these publications.

To acknowledge the time and effort you spent reviewing our manuscript, we agreed to include some of the information you had asked for as short summaries. Nevertheless, for the sake of consistency, and in line with the guidelines of the International Committee of Medical Journal Editors (ICMJE) regarding overlapping publications (https://www.icmje.org/recommendations/browse/publishing-and-editorial-issues/overlapping-publications.html), we do not agree on the addition of further, indirectly related data to this manuscript.

We understand that you will certainly find our answer unsuitable and are aware that you will certainly recommend that our manuscript be rejected. While we would certainly regret this decision, we would rather have our manuscript rejected than compromise with its consistency or with our ethics.

We believe that an Editorial Decision is now required to sort out our disagreement and make a final decision on our manuscript.

We wish you all the best and thank you again for your time.